# Is latent tuberculosis infection challenging in Iranian health care workers? A systematic review and meta-analysis

Mohammad Hossein YektaKooshali[1,2], Farahnaz Movahedzadeh[3,4], Ali Alavi Foumani[2], Hoda Sabati[5], Alireza Jafari[2,6]*

1 Student Research Committee, School of nursing, Midwifery and Paramedicine, Guilan University of Medical Sciences, Rasht, Iran, 2 Inflammatory Lung Diseases Research Center, Department of Internal Medicine, Razi Hospital, School of Medicine, Guilan University of Medical Sciences, Rasht, Iran, 3 Institute for Tuberculosis Research, College of Pharmacy, University of Illinois at Chicago, Chicago, Illinois, United States of America, 4 Department of pharmaceutical Sciences, College of Pharmacy University of Illinois at Chicago, Chicago, Illinois, United States of America, 5 Biotechnology and Biological Science Research Center, Faculty of Science, Shahid Chamran University of Ahvaz, Ahvaz, Iran, 6 Urology Research Center, Department of Internal Medicine, Razi Hospital, School of Medicine, Guilan University of Medical Sciences, Rasht, Iran

* Dr.alireza.jafariii@gmail.com

**Data Availability Statement:** All relevant data are within the manuscript and its Supporting Information files.

## Abstract

### Background

The high chances of getting latent tuberculosis infection (LTBI) among health care workers (HCWs) will an enormous problem in low and upper-middle-income countries.

### Method

Search strategies were done through both national and international databases include SID, Barakat knowledge network system, Irandoc, Magiran, Iranian national library, web of science, Scopus, PubMed/MEDLINE, OVID, EMBASE, the Cochrane library, and Google Scholar search engine. The Persian and the English languages were used as the filter in national and international databases, respectively. Medical Subject Headings (MeSH) terms was used to controlling comprehensive vocabulary. The search terms were conducted without time limitation till January 01, 2019.

### Results

The prevalence of LTBI in Iranian's HCWs, based on the PPD test was 27.13% [CI95%: 18.64–37.7]. The highest prevalence of LTBI in Iranian's HCWs were estimated 41.4% [CI95%: 25.4–59.5] in the north, and 33.8% [CI95%: 21.1–49.3] in the west. The lowest prevalence of LTBI was evaluated 18.2% [CI95%: 3.4–58.2] in the south of Iran. The prevalence of LTBI in Iranian's HCWs who had work-experience more than 20 years old were estimated 20.49% [CI95%: 11–34.97]. In the PPD test, the prevalence of LTBI in Iranian's HCWs who had received the Bacille Calmette–Guérin (BCG) was estimated 15% [CI95%: 3.6–47.73]. While, in the QFT, the prevalence of LTBI in Iranian's HCWs in non-vaccinated was estimated 25.71% [CI95%: 13.96–42.49].

**Funding:** The author(s) received no specific funding for this work.

**Competing interests:** The authors have declared that no competing interests exist.

## Conclusions

This meta-analysis shows the highest prevalence of LTBI in Iranian's HCWs in the north and the west probably due to neighboring countries like Azerbaijan and Iraq, respectively. It seems that Iranian's HCWs have not received the necessary training to prevent of TB. We also found that BCG was not able to protect Iranian's HCWs from TB infections, completely.

## 1. Introduction

Latent tuberculosis infection (LTBI) is an immune response to *Mycobacterium tuberculosis* (Mtb) antigens without symptoms of active tuberculosis (TB) [1]. Mtb is able to colonize inside the alveolar macrophages and finally form granuloma. Mtb is ingested by phagocytosis by resident alveolar macrophages and tissue dendritic cells (DC) [2, 3]. The immune cells contribute and the pathological mark of TB, the granuloma, is formed. In the granuloma, macrophages differentiate into epithelial cells or foamy macrophages, or fuse to form giant cells, and become surrounded by lymphocytes, fibroblasts and extracellular matrix proteins. In such conditions, the Mtb will be surviving until the granuloma fails due to immunosuppression [4, 5]. Mtb use the granuloma as they are effective at initial infection level since they recruit new macrophages to allow the spread of infection between host cells [6]. At this stage, the LTBI is formed in the patient's body [7].

There are several reports of TB outbreaks in Iran. According to the Iranian's ministry of health, the incidence and the prevalence of TB are high in Sistan and Baluchestan, Khorasan, Mazandaran, Guilan, West and East Azerbaijan, Ardabil, Kurdistan, Khuzestan and southern coasts. Conversely, the incidence and the prevalence of TB are low in the central parts of Iran. The highest incidence and prevalence of TB belong to Golestan and Sistan-Baluchistan [8].

The risk of tuberculosis in health care workers (HCWs) is estimated to be twice as high in the general population, in high-income countries, and five times higher than the general population in countries with a low and middle income [9, 10]. In addition, one of the challenges in many countries is the transfer of tuberculosis from patients admitted to the hospital to HCWs [10]. Most importantly, the transfer of resistant Mycobacterium tuberculosis strains from admitted patients to HCWs has increased the importance of the subject [11].

According to the findings, direct exposure to HCWs in patients with tuberculosis, direct contact with phlegm specimens and blood products of suspected tuberculosis patients, and long hours of work in high-risk places increases the risk of tuberculosis infection [12, 13]. This means that direct contact is one of the most important and worrisome factor in the transmission of tuberculosis to HCWs [10–12, 14]. Work experience, age [15], occupational status [16], the use of personal protective equipment, ventilation [17], hospital infection control unit and infection control in isolation rooms can affect LTBI outbreaks in HCWs [9–12]. To diagnose LTBI, the mantoux tuberculin skin test (TST) and QuantiFERON-TB Gold (QFT) are used [18]. Studies have shown that QFT has a higher sensitivity and specificity in detecting LTBI [19, 20]. However, some researchers believe that QFT is not superior to TST in detecting LTBI [21–23].

The early detection of LTBI in controlling, treating and preventing Mtb is a key element in patients who preventive treatment can reduce the risk of active tuberculosis in patients by up to 90% [24]. So far, systematic review and meta-analysis has not been conducted to evaluate the prevalence and risk factors of LTBI among Iranian's HCWs. In Iran, the Centers for Disease Control and Prevention (CDC) do not control the Mtb as a regular program, however, reports of LTBI outbreaks in HCWs attracts a high controversy [25]. Due to the highest level

of evidence and an essential role in evidence-based decision-making of meta-analysis studies [26, 27]. This study estimated the prevalence and risk factors of LTBI among Iranian's HCWs which can have vital information for policy-makers and planning at the country level.

## 2. Methods

### 2.1. Study protocol

This is the first study that was conducted based on the meta-analysis of observational studies according to epidemiology guidelines [27], and the PRISMA (Preferred Reporting Items for Systematic Reviews and Meta-Analyses) statement (S1 File) [28]. The study was achieved based on five steps; design and search strategy; collecting original articles; evaluating inclusion and exclusion criteria, and finally qualitative evaluation and statistical analysis of data. Two independent researchers (MH.YK & A.J) evaluated the data. The disagreements were solved by consensus between the team and a Bacteriologist (H.S.E). The review protocol was registered in International Prospective Register of Systematic Reviews (PROSPERO) (https://www.crd.york.ac.uk/PROSPERO/) Identifier: CRD42018117682 [29, 30] (S2 File).

### 2.2. Search strategy

In order to maximize its sensitivity, search strategy was lead through Persian (national) databases, include scientific information database (http://www.sid.ir), Barakat knowledge network system (http://health.barakatkns.com), Iranian research institute for information science and technology (https://irandoc.ac.ir), MagIran (http://www.magiran.com), Iranian national library (http://www.nlai.ir/). The international databases, including web of science, Scopus, PubMed/MEDLINE, OVID, EMBASE, the Cochrane Library (Cochrane Database of Systematic Reviews), and Google Scholar search engine. The Persian and the English languages were used as the filter in national and international databases, respectively. The search terms were adapted to international databases. To search a combination of words, Boolean operators (AND & OR) were used. Searching was done through medical subject heading (MeSH) terms. The search terms were conducted without any time limitation till January 01, 2019. The authors independently analyzed the manuscript contained in the title and abstract. For instance, PubMed search formula was provided in the appendix.

### 2.3. Inclusion and exclusion criteria

**2.3.1. Inclusion criteria based on PICO (related to evidence-based medicine) [31, 32].** Inclusion criteria were determined based on PICO model. In this study, population was the population of Iranian's HCWs who were residents in the geographic regions of Northern, Southern, Eastern, Western of Iran. Comparison was conducted on a population of HCWs who did not have signs of active TB disease and did not feel illness. Outcome was the overall prevalence of LTBI infection among Iranian's HCWs.

**2.3.2. Exclusion criteria.** In this study, review articles, letters, editorial, case reports, conference papers, and comments were excluded. The studies which did not have a focus on the prevalence of LTBI in Iranian's HCWs, duplicated papers, non-English full papers, non-Persian full papers, and non-accessible full-text papers were excluded. Likewise, the populations other than Iranian's HCWs were excluded.

### 2.4. Latent TB detection criteria

**2.4.1. The Mantoux tuberculin skin test (TST).** To the Mantoux tuberculin skin test (TST), purified protein derivative (0.1 Ml) is used [33–35], and the induration at TST site is

measured 72 hours later. TST reaction of $\geq 5$ mm of induration is classified as negative but is considered as positive in patients receiving corticosteroid or patients with Acquired Immuno-deficiency Syndrome (AIDS), diabetes mellitus, lymphoma, and leukemia. The induration of $\geq 10$ mm is classified positive in; recent immigrants ($< 5$ years) from high-prevalence countries; residents and employees of high-risk congregate settings; mycobacteriology laboratory personnel; persons with clinical conditions that place them at high risk. The induration of $\geq 15$ mm is considered positive in any person, including persons with no known risk factors for TB. Two-step testing methods were used for health care workers and nursing home residents [33–35].

**2.4.2. Interferon-gamma release assays.** Interferon-gamma release assays (IGRAs) show how the immune system reacts to the Mycobacteria that cause TB [36]. The IGRA has been approved by the U.S. Food and Drug Administration (FDA). Positive IGRA means that the person has been infected with TB bacteria. Negative IGRA means that the person's blood did not react to the test and that latent TB infection or TB disease is not likely. IGRA is the preferred method of TB infection testing for people who have received the Bacille Calmette–Guérin (BCG) [30, 36–39].

## 2.5. Selection of studies

During the selection stage, duplicated studies were removed by the EndNote™ software Ver. X9 (Clarivate Analytics company). In the skimming and screening stage, co-authors, journals, and publishing years were evaluated by two experts based on inclusion and exclusion criteria (the eligibility stage), independently. The disagreements between the two were resolved through an expert bacteriologist (Fig 1).

## 2.6. Quality appraisal

In this stage, the irrelevant studies were excluded, and then the quality of each study was evaluated. To quality appraisal, the Newcastle-Ottawa Scale (NOS) checklist (S3 File) [40] was applied which is determined the quality of these studies based on three levels of scoring. The score of five or less defined a poor quality study; the score of five or six distinguished as the medium quality study, and the score of seven or eight determined as the high-quality study. Finally, the medium to high-quality studies were included in the data analysis (Fig 1).

## 2.7. Data extraction

The enter terms were author's names, province, geographical regions, year of publishing, sample size, age, gender, history of BCG, history of exposure with tuberculosis, history of tuberculosis disease, laboratory diagnosis tests, job experience, duration of employment, workplaces, single-step or two-step TST, and history of hospitalization. The author's name, institution, and the journal name were blinded, and then data was extracted through two researchers (MH.YK & A.J), independently. Only if necessary, the additional information/raw data was collected by phone call, mailing or fax.

## 2.8. Statistical analysis

The prevalence of LTBI in HCWs was considered as a binomial distribution probability, and the variance was calculated by a binomial distribution. To evaluate its heterogeneity, the Cochran test (Q) and $I^2$ index were used [1, 41, 42]. The subgroup analysis was performed based on province, single-step or two-step TST, laboratory diagnosis tests, job, gender, history of TB disease, history of TB exposure, history of BCG, and geographical region. Sensitivity

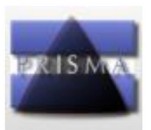

**PRISMA 2009 Flow Diagram**

Records identified through database searching
(n =309)

Additional records identified through other sources
(n = 112 )

Records after duplicates removed
(n = 212 )

Records screened
(n = 212)

Records excluded due to skimming the title & abs and irrelevancy
(n =151)

Full-text articles assessed for eligibility
(n =61)

Full-text articles excluded, with reasons
(n =41), with reasons:
- Non-Iranian Sample size (n=23)
- No full text found (n=2)
- Controlling Sample size (n=15)
- Low quality (n=1)

Studies included in qualitative synthesis
(n =20)

Studies included in quantitative synthesis (meta-analysis)
(n = 20)

*Identification*

*Screening*

*Eligibility*

*Included*

**Fig 1. A flow diagram following the PRISMA (depicted by MH-YK).** *From*: Moher D, Liberati A, Tetzlaff J, Altman DG, The PRISMA Group (2009). *Preferred Reporting Items for Systematic Reviews and Meta-Analyses: The PRISMA Statement. PLoS Med 6(7): e1000097. doi:10.1371/journal. pmed1000097* **For more information, visit** www.prisma-statement.org.

analysis was also achieved to evaluate the impact of each study, based on the results of the overall prevalence of LTBI in Iranian HCWs. The Begg's test and Egger's test were carried out using a funnel plot to examine publication bias. Data analysis was examined by the comprehensive meta-analysis (Ver. 2 Englewood, NJ 07631, USA), and the level of significance was considered as p<0.05.

## 3. Results

### 3.1. Study characteristics and methodological quality

In the primary search of study, 421 studies were found. After skimming and screening, 20 (4.75%) studies were eligible according to inclusions and exclusions criteria [43–62]. The total sample size was calculated 6453 Iranian's HCWs (Fig 1) (S1 Table).

### 3.2. The overall prevalence LTBI in HCWs

The prevalence of LTBI in HCWs, based on the PPD test (48 hours) was 27.13% [CI95%: 18.64–37.7] (Fig 2), and based on the QFT test was 16.92% [CI95%: 9.7–27.84] (Fig 3). The prevalence of LTBI was estimated 12.11% [CI95%: 4.53–28.57] in Iranian's HCWs who had negative TST reaction (48 hours) in the first week (Fig 4). The prevalence of induration at TST site (48 h) was estimated <4 mm in 43.74% [CI 95%: 28.19–60.63], 5–9 mm in 17.52% [CI 95%: 9.73–29.5], 10–15 mm in 14.55% [CI 95%: 8.87–22.93] and >15 mm in 13.4% [CI 95%: 8.59–20.31] (S1 Fig).

### 3.3. The prevalence of LTBI in Iranian HCWs based on geographical region of Iran

The highest prevalence of LTBI in Iranian's HCWs was estimated 41.4% [CL95%: 25.4–59.5] in the north, and 33.8% [CI95%: 21.1–49.3] in the west of Iran. The lowest prevalence of LTBI was found 18.2% [CI95%: 3.4–58.2] in the south of Iran. These results showed a significant relationship between LTBI prevalence in Iranian's HCWs and the geographic location in Iran (p <0.0001) (S2 Fig) (Fig 5).

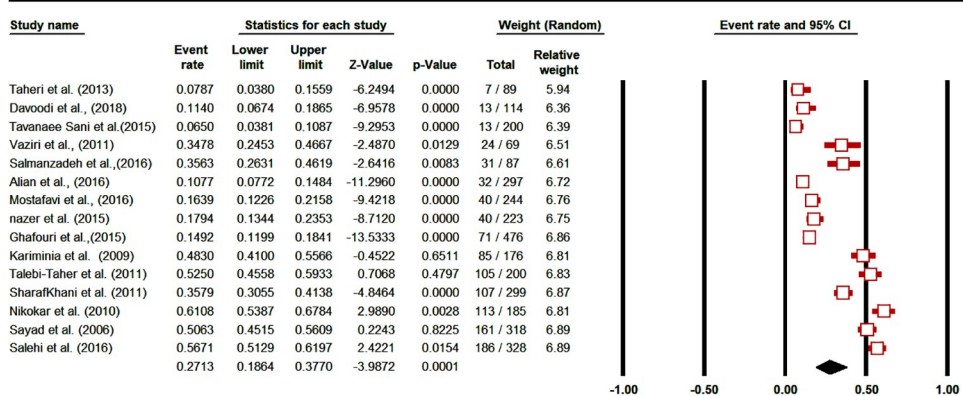

**Meta Analysis**

**Fig 2. The prevalence subgroup analysis based on TST/PPD induration diameter (48 hrs.) in Iranian's HCWs with LTBI (forest plot-random effect model).**

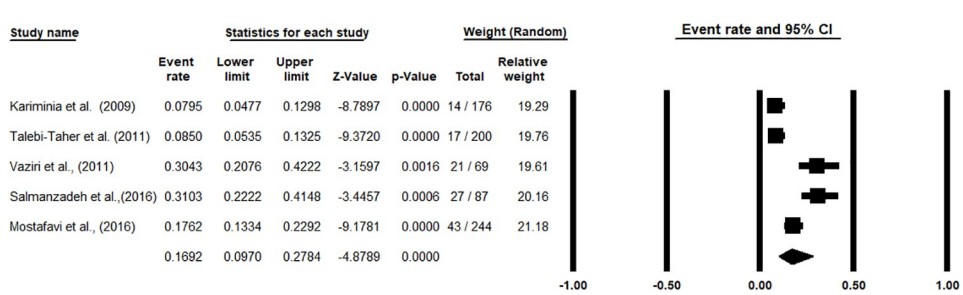

**Meta Analysis**

**Fig 3. The prevalence subgroup analysis based on QFT in Iranian's HCWs with LTBI (forest plot—Random effect model).**

## 3.4. Sensitivity analysis and cumulative meta-analysis

Sensitivity analysis of prevalence of LTBI in Iranian's HCWs was estimated with a 95% confidence interval. It showed that there is no significant effect on the overall prevalence of LTBI in Iranian's HCWs (Fig 6). The overall prevalence of LTBI in Iranian's HCWs based on the publication year was estimated by cumulative meta-analysis and represented in (S3 Fig). The sub group analysis of the quality of studies was showed in (S4 Fig).

## 3.5. Meta-regression

Base on the PPD (48 hours) results, the prevalence of publishing manuscripts about identification of LTBI in Iranian's HCWs has decreased in Iran. There was no significant relationship between publishing years. (Mixed effects regression (Method of moments); Slope = -0.1898 (SE = 0.068, (95% CI: -0.323– -0.056)), Intercept = 381.14 (SE = 137.43, (95% CI: 111.78–650.5)), P = 0.10653) (Fig 7).

## 3.6. The prevalence of LTBI in HCWs based on term of employment

Base on the PPD results, the prevalence of LTBI in Iranian's HCWs with more than 10 years old work-experience was evaluated 51%. The prevalence of LTBI in Iranian's HCWs with less than 10 years old work-experience was estimated at 29.30%.

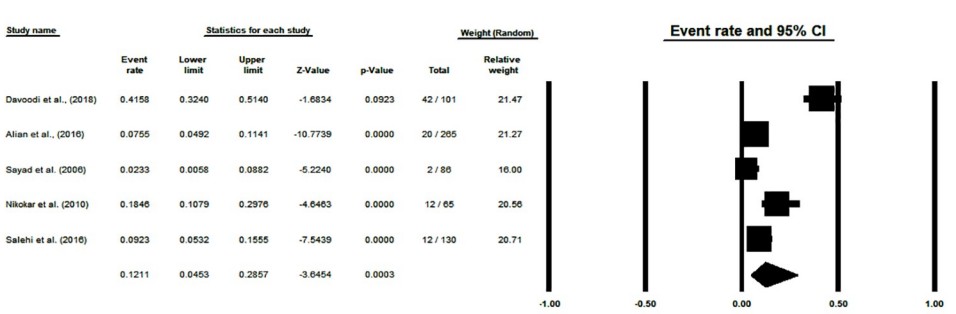

**Meta Analysis**

**Fig 4. The prevalence subgroup analysis based on TST/PPD induration diameter after one week in Iranian's HCWs with LTBI (forest plot—Random effect model).**

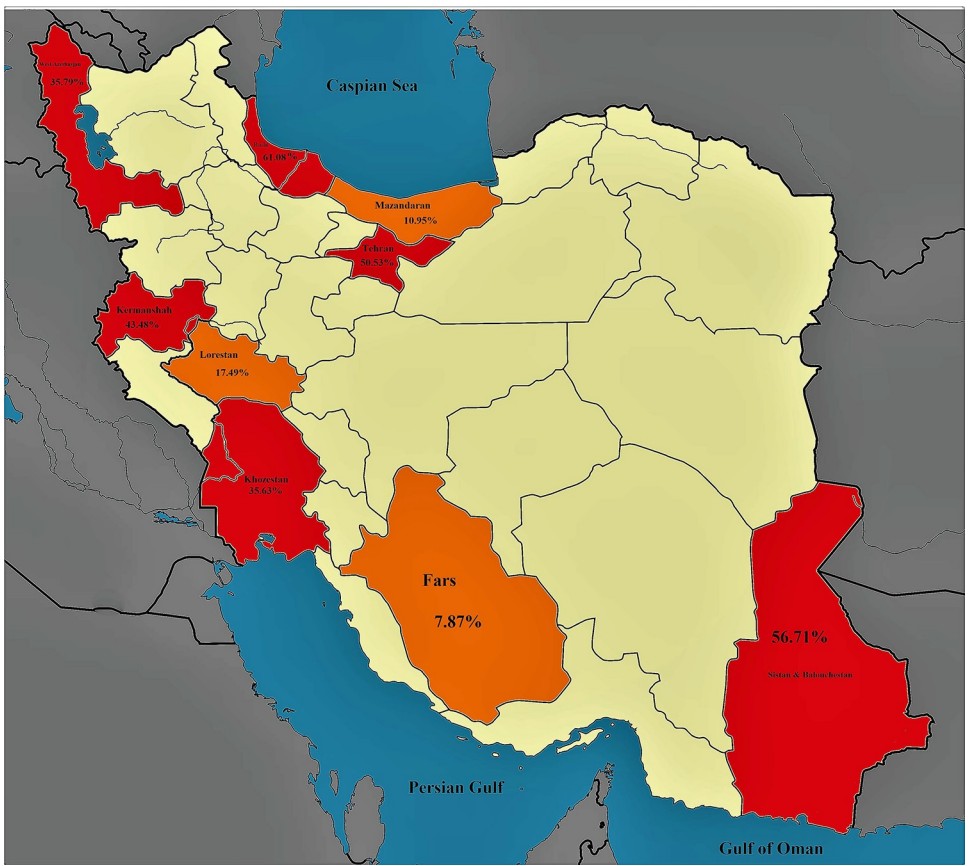

**Fig 5. Demonstrating LTBI in Iranian's HCWs based on geographical classification (random effect model).**

Base on the QFT results, the prevalence of LTBI in Iranian's HCWs with more than 20 years old work-experience was calculated 20.49% [CI95%: 11–34.97], which showed a significant relationship between the duration of employment (P <0.0001) (S5 Fig).

### 3.7. The prevalence of LTBI in Iranian HCWs based on occupation and wards

Base on the PPD results, the prevalence of LTBI in assistant nurses was estimated 45.76% [CI 95%: 33.51–58.55], in physicians was estimated 44.99% [CI95%: 33.37–57.17], in ward nurses was calculated 39.4% [CI95%: 17.63–66.39], and in service workers was estimated 36.43% [CI95%: 19.51–57.53]. Base on the QFT results, the prevalence of LTBI in both nurses and TB service workers was higher than other occupations (Fig 8).

The prevalence of LTBI in the infectious ward was estimated 52.09% [CI95%: 43.92–60.14], and in the internal ward was evaluated 50% [CI95%: 34.22–65.78]. The lowest prevalence of LTBI was estimated in the infectious wards based on QFT. There was a significant relationship between the prevalence of LTBI in Iranian's HCWs, and hospital wards (p <0.0001) (Fig 9).

### 3.8. The prevalence of LTBI in Iranian's HCWs based on gender and age

The prevalence of LTBI was estimated at 42.16% [CI95%: 26.41–59.69] in male Iranian's HCWs based on the PPD test. The prevalence of LTBI in Iranian's HCWs and type of gender

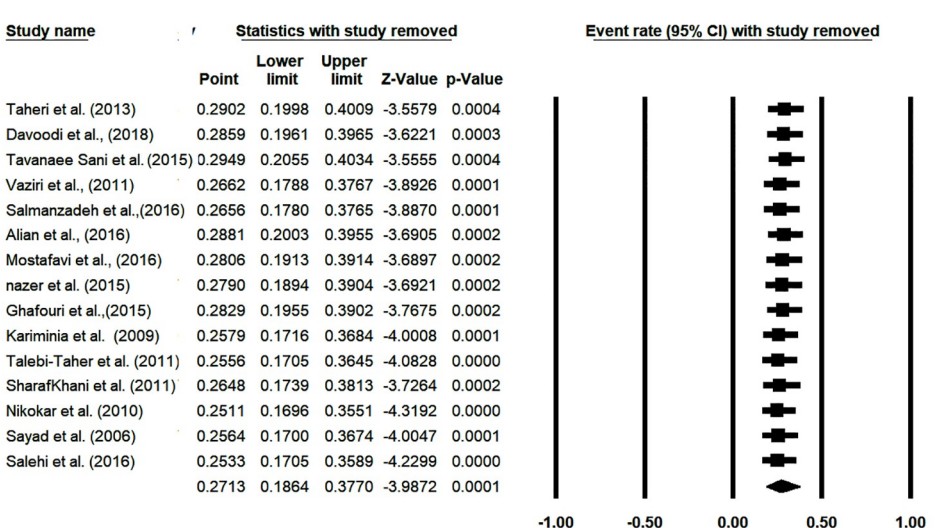

| Study name | | Statistics with study removed | | | | Event rate (95% CI) with study removed |
|---|---|---|---|---|---|---|
| | Point | Lower limit | Upper limit | Z-Value | p-Value | |
| Taheri et al. (2013) | 0.2902 | 0.1998 | 0.4009 | -3.5579 | 0.0004 | |
| Davoodi et al., (2018) | 0.2859 | 0.1961 | 0.3965 | -3.6221 | 0.0003 | |
| Tavanaee Sani et al. (2015) | 0.2949 | 0.2055 | 0.4034 | -3.5555 | 0.0004 | |
| Vaziri et al., (2011) | 0.2662 | 0.1788 | 0.3767 | -3.8926 | 0.0001 | |
| Salmanzadeh et al.,(2016) | 0.2656 | 0.1780 | 0.3765 | -3.8870 | 0.0001 | |
| Alian et al., (2016) | 0.2881 | 0.2003 | 0.3955 | -3.6905 | 0.0002 | |
| Mostafavi et al., (2016) | 0.2806 | 0.1913 | 0.3914 | -3.6897 | 0.0002 | |
| nazer et al. (2015) | 0.2790 | 0.1894 | 0.3904 | -3.6921 | 0.0002 | |
| Ghafouri et al.,(2015) | 0.2829 | 0.1955 | 0.3902 | -3.7675 | 0.0002 | |
| Kariminia et al. (2009) | 0.2579 | 0.1716 | 0.3684 | -4.0008 | 0.0001 | |
| Talebi-Taher et al. (2011) | 0.2556 | 0.1705 | 0.3645 | -4.0828 | 0.0000 | |
| SharafKhani et al. (2011) | 0.2648 | 0.1739 | 0.3813 | -3.7264 | 0.0002 | |
| Nikokar et al. (2010) | 0.2511 | 0.1696 | 0.3551 | -4.3192 | 0.0000 | |
| Sayad et al. (2006) | 0.2564 | 0.1700 | 0.3674 | -4.0047 | 0.0001 | |
| Salehi et al. (2016) | 0.2533 | 0.1705 | 0.3589 | -4.2299 | 0.0000 | |
| | 0.2713 | 0.1864 | 0.3770 | -3.9872 | 0.0001 | |

**Meta Analysis**

**Fig 6. Sensitivity analysis to prevalence of LTBI in Iranian's HCWs (one study removed test).**

based on PPD test (P<0.051). In QFT, however, a significant relationship was showed between the prevalence of LTBI in Iranian's HCWs, and the gender (P <0.0001) (S6 Fig).

The highest prevalence of LTBI in Iranian's HCWs who were more than 40 years old was estimated 44% [CI 95%: 26.47–63.16] in the PPD test.

The highest prevalence of LTBI in Iranian's HCWs aged 30 years old was estimated 22.52% [CI95%: 3.7–68.34] in the QFT. In both PPD test and QFT, it was evaluated that there was significant relationship between the prevalence of LTBI in Iranian's HCWs, and age of HCWs (P<0.0001) (S7 Fig).

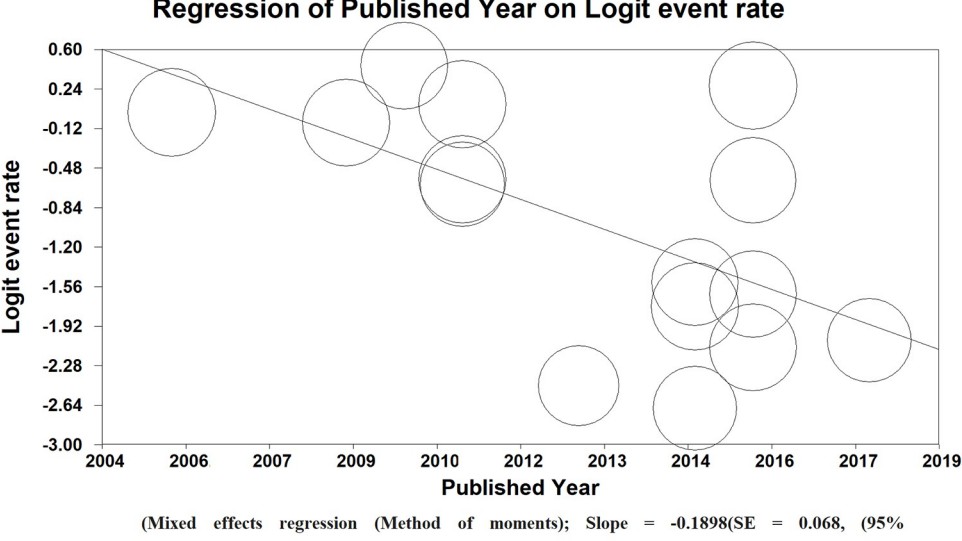

**Regression of Published Year on Logit event rate**

(Mixed effects regression (Method of moments); Slope = -0.1898(SE = 0.068, (95% CI: -0.323– -0.056)), Intercept = 381.14(SE = 137.43, (95% CI: 111.78–650.5)), P = 0.10653)

**Fig 7. Meta-regression of LTBI in Iranian's HCWs according to publishing year of studies (method of moments).**

**(A)**

| Group by Subgroup within study Occupation | Study name | Statistics for each study | | | | | | Relative weight | Event rate and 95% CI |
|---|---|---|---|---|---|---|---|---|---|
| | | Event rate | Lower limit | Upper limit | Z-Value | p-Value | Total | | |
| Administrative | Mostafavi et al., (2016)@ | 0.1905 | 0.0982 | 0.3370 | -3.6822 | 0.0002 | 8 / 42 | 40.30 | |
| Administrative | Hashemi et al. (2014)@ | 0.3000 | 0.0998 | 0.6237 | -1.2279 | 0.2195 | 3 / 10 | 27.08 | |
| Administrative | Talebi-Taher et al. (2011)@ | 0.5385 | 0.2817 | 0.7764 | 0.2771 | 0.7817 | 7 / 13 | 32.62 | |
| Administrative | | 0.3182 | 0.1452 | 0.5617 | -1.4788 | 0.1392 | | | |
| Assistant Nurse | Rahbar et al. (2007)* | 0.3400 | 0.2230 | 0.4805 | -2.2218 | 0.0263 | 17 / 50 | 31.64 | |
| Assistant Nurse | Talebi-Taher et al. (2011)* | 0.5814 | 0.4311 | 0.7180 | 1.0627 | 0.2879 | 25 / 43 | 30.74 | |
| Assistant Nurse | Hashemi et al. (2014)* | 0.4595 | 0.3498 | 0.5731 | -0.6967 | 0.4860 | 34 / 74 | 37.62 | |
| Assistant Nurse | | 0.4576 | 0.3351 | 0.5855 | -0.6468 | 0.5178 | | | |
| Finance staff | Mostafavi et al., (2016)! | 0.1000 | 0.0139 | 0.4672 | -2.0845 | 0.0371 | 1 / 10 | 100.00 | |
| Finance staff | | 0.1000 | 0.0139 | 0.4672 | -2.0845 | 0.0371 | | | |
| Intern Student | Alian et al., (2016)# | 0.2533 | 0.1678 | 0.3634 | -4.0713 | 0.0000 | 19 / 75 | 100.00 | |
| Intern Student | | 0.2533 | 0.1678 | 0.3634 | -4.0713 | 0.0000 | | | |
| No TB Service worker | Rahbar et al. (2007)> | 0.2222 | 0.1240 | 0.3659 | -3.4938 | 0.0005 | 10 / 45 | 35.45 | |
| No TB Service worker | nazer et al. (2015)> | 0.3684 | 0.1868 | 0.5970 | -1.1333 | 0.2571 | 7 / 19 | 30.05 | |
| No TB Service worker | Talebi-Taher et al. (2011)> | 0.5357 | 0.3544 | 0.7080 | 0.3776 | 0.7057 | 15 / 28 | 34.50 | |
| No TB Service worker | | 0.3643 | 0.1951 | 0.5753 | -1.2682 | 0.2047 | | | |
| Nurse | nazer et al. (2015)- | 0.1823 | 0.1326 | 0.2454 | -7.7955 | 0.000033 | / 181 | 20.78 | |
| Nurse | Rahbar et al. (2007)- | 0.2733 | 0.2080 | 0.3501 | -5.3370 | 0.000041 | / 150 | 21.04 | |
| Nurse | Hashemi et al. (2014)- | 0.2679 | 0.1941 | 0.3573 | -4.7125 | 0.000030 | / 112 | 20.19 | |
| Nurse | Vaziri et al., (2011)- | 0.3478 | 0.2453 | 0.4667 | -2.4870 | 0.0129 | 24 / 69 | 19.01 | |
| Nurse | Talebi-Taher et al. (2011)- | 0.5161 | 0.3933 | 0.6371 | 0.2540 | 0.7995 | 32 / 62 | 18.97 | |
| Nurse | | 0.3041 | 0.2129 | 0.4138 | -3.3826 | 0.0007 | | | |
| other Low risk stuff | Mostafavi et al., (2016)" | 0.1412 | 0.0820 | 0.2324 | -5.7963 | 0.0000 | 12 / 85 | 30.29 | |
| other Low risk stuff | SharafKhani et al. (2011)" | 0.3371 | 0.2710 | 0.4103 | -4.2278 | 0.000059 | / 175 | 34.86 | |
| other Low risk stuff | Kariminia et al. (2009)" | 0.4808 | 0.4035 | 0.5590 | -0.4803 | 0.631075 | / 156 | 34.85 | |
| other Low risk stuff | | 0.3080 | 0.1672 | 0.4967 | -1.9928 | 0.0463 | | | |
| Paramedics | Rahbar et al. (2007) | 0.2444 | 0.1408 | 0.3897 | -3.2533 | 0.0011 | 11 / 45 | 100.00 | |
| Paramedics | | 0.2444 | 0.1408 | 0.3897 | -3.2533 | 0.0011 | | | |
| Physician | Rahbar et al. (2007)% | 0.3333 | 0.1460 | 0.5940 | -1.2655 | 0.2057 | 5 / 15 | 20.54 | |
| Physician | Talebi-Taher et al. (2011)% | 0.4815 | 0.3525 | 0.6129 | -0.2721 | 0.7855 | 26 / 54 | 79.46 | |
| Physician | | 0.4499 | 0.3337 | 0.5717 | -0.8047 | 0.4210 | | | |
| Physiopatthology Student | Alian et al., (2016)$ | 0.1391 | 0.0925 | 0.2039 | -7.7514 | 0.000021 | / 151 | 100.00 | |
| Physiopatthology Student | | 0.1391 | 0.0925 | 0.2039 | -7.7514 | 0.0000 | | | |
| TB lab staff | Nikokar et al. (2010)< | 0.1667 | 0.0921 | 0.2828 | -4.6460 | 0.0000 | 10 / 60 | 51.54 | |
| TB lab staff | Mostafavi et al., (2016)/ | 0.5000 | 0.2939 | 0.7061 | 0.0000 | 1.0000 | 10 / 20 | 48.46 | |
| TB lab staff | | 0.3037 | 0.0827 | 0.6785 | -1.0314 | 0.3024 | | | |
| TB Service worker | Mostafavi et al., (2016)< | 0.0313 | 0.0044 | 0.1911 | -3.3799 | 0.0007 | 1 / 32 | 100.00 | |
| TB Service worker | | 0.0313 | 0.0044 | 0.1911 | -3.3799 | 0.0007 | | | |
| Ward sister | Rahbar et al. (2007)^ | 0.2667 | 0.1581 | 0.4132 | -3.0009 | 0.0027 | 12 / 45 | 48.77 | |
| Ward sister | Hashemi et al. (2014)^ | 0.5306 | 0.3921 | 0.6646 | 0.4283 | 0.6684 | 26 / 49 | 51.23 | |
| Ward sister | | 0.3940 | 0.1763 | 0.6639 | -0.7594 | 0.4476 | | | |
| Overall | | 0.2566 | 0.3343 | | -9.1609 | 0.0000 | | | |

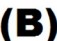

-1.00    -0.50    0.00    0.50    1.00

**(B)**

| Group by Subgroup within study Occupation | Study name | Statistics for each study | | | | | | Relative weight | Event rate and 95% CI |
|---|---|---|---|---|---|---|---|---|---|
| | | Event rate | Lower limit | Upper limit | Z-Value | p-Value | Total | | |
| TB Service worker | Mostafavi et al., (2016)\ | 0.2188 | 0.1080 | 0.3930 | -2.9769 | 0.0029 | 7 / 32 | 100.00 | |
| TB Service worker | | 0.2188 | 0.1080 | 0.3930 | -2.9769 | 0.0029 | | | |
| Administrative | Talebi-Taher et al. (2011)< | 0.0769 | 0.0107 | 0.3906 | -2.3874 | 0.0170 | 1 / 13 | 15.22 | |
| Administrative | Mostafavi et al., (2016)< | 0.1429 | 0.0656 | 0.2834 | -4.0633 | 0.0000 | 6 / 42 | 84.78 | |
| Administrative | | 0.1304 | 0.0634 | 0.2495 | -4.6727 | 0.0000 | | | |
| Assistant Nurse | Talebi-Taher et al. (2011)> | 0.1163 | 0.0492 | 0.2505 | -4.2633 | 0.0000 | 5 / 43 | 100.00 | |
| Assistant Nurse | | 0.1163 | 0.0492 | 0.2505 | -4.2633 | 0.0000 | | | |
| No TB Service worker | Talebi-Taher et al. (2011)\ | 0.1429 | 0.0547 | 0.3245 | -3.3177 | 0.0009 | 4 / 28 | 100.00 | |
| No TB Service worker | | 0.1429 | 0.0547 | 0.3245 | -3.3177 | 0.0009 | | | |
| Nurse | Talebi-Taher et al. (2011)* | 0.0806 | 0.0340 | 0.1795 | -5.2177 | 0.0000 | 5 / 62 | 48.68 | |
| Nurse | Vaziri et al., (2011)* | 0.5072 | 0.3910 | 0.6227 | 0.1204 | 0.9042 | 35 / 69 | 51.32 | |
| Nurse | | 0.2369 | 0.0271 | 0.7760 | -0.9505 | 0.3419 | | | |
| other Low risk stuff | Kariminia et al. (2009)# | 0.0705 | 0.0395 | 0.1228 | -8.2460 | 0.000011 | / 156 | 49.17 | |
| other Low risk stuff | Mostafavi et al., (2016)# | 0.1882 | 0.1186 | 0.2854 | -5.2672 | 0.0000 | 16 / 85 | 50.83 | |
| other Low risk stuff | | 0.1181 | 0.0429 | 0.2858 | -3.5999 | 0.0003 | | | |
| Physician | Talebi-Taher et al. (2011)^ | 0.0370 | 0.0093 | 0.1364 | -4.5215 | 0.0000 | 2 / 54 | 100.00 | |
| Physician | | 0.0370 | 0.0093 | 0.1364 | -4.5215 | 0.0000 | | | |
| TB lab staff | Kariminia et al. (2009)/ | 0.1500 | 0.0492 | 0.3758 | -2.7699 | 0.0056 | 3 / 20 | 23.43 | |
| TB lab staff | Mostafavi et al., (2016)/ | 0.1667 | 0.0921 | 0.2828 | -4.6460 | 0.0000 | 10 / 60 | 76.57 | |
| TB lab staff | | 0.1626 | 0.0968 | 0.2603 | -5.4063 | 0.0000 | | | |
| Overall | | 0.1066 | 0.1864 | | -10.8086 | 0.0000 | | | |

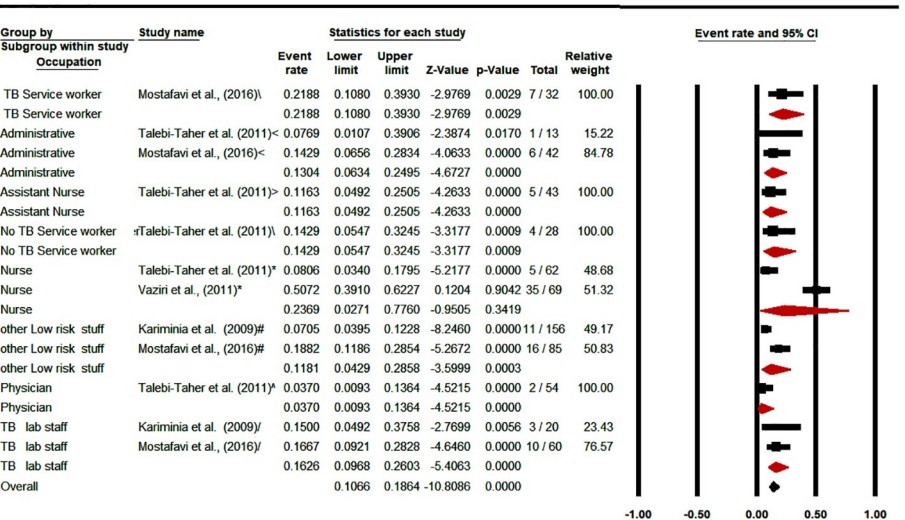

-1.00    -0.50    0.00    0.50    1.00

**Fig 8. The prevalence subgroup analysis of occupational based on PPD (A), and QFT (B) in Iranian's HCWs with LTBI (forest plot—Random effect model).**

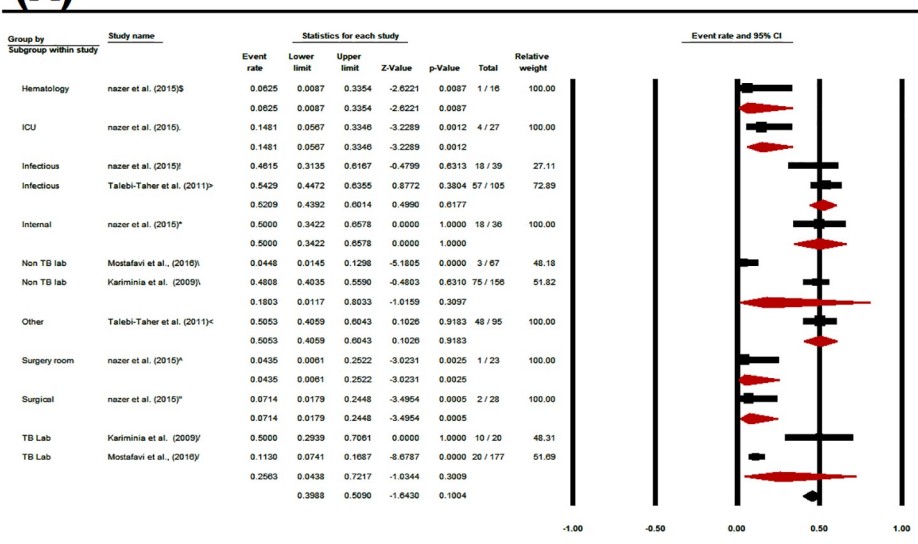

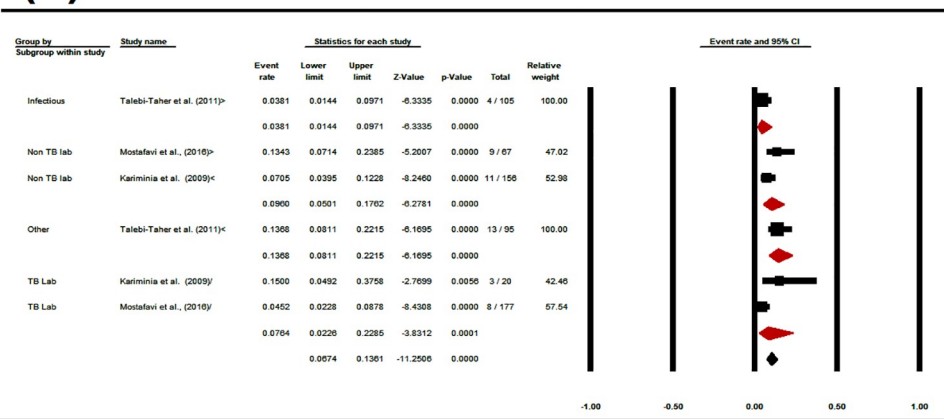

**Fig 9. The prevalence subgroup analysis of ward based on PPD (A), and QFT (B) in Iranian's HCWs with LTBI (forest plot—Random effect model).**

### 3.9. The prevalence of LTBI in Iranian's HCWs based on the history of tuberculosis contact and the tuberculosis clinical symptoms

The results showed that 30.15% [CI 95%: 11–60.13] of Iranian's HCWs directly contacted to patients with tuberculosis. The results also showed that 6.9% [CI 95%: 2.36–18.55] of Iranian's HCWs had active tuberculosis symptoms (S8 and S9 Figs).

### 3.10. The prevalence of LTBI in Iranian's HCWs based on the "BCG"

The prevalence of LTBI in Iranian's HCWs who received the BCG was estimated 15% [CI 95%: 3.6–47.73] based on the PPD test. While the prevalence of LTBI in Iranian's HCWs who did not receive the BCG was estimated at 25.71% [CI95%: 13.96–42.49] based on the QFT. In both PPD, and QFT, there was a significant relationship between those who did and those who did not receive the vaccine (P<0.0001) (S10 Fig).

### 3.11. Publication bias

The publication bias in this study was evaluated by Begg's and Egger's tests. The publication bias by Begg's test was calculated 0.06, and the Egger's test was calculated 0.028. The probability of the publication bias in this study was significant (S11 Fig).

## 4. Discussion

This study is the first systematic review and meta-analysis of LTBI outbreak been carried out in Iranian's HCWs. According to results of the current meta-analysis, the prevalence of LTBI in Iranian's HCWs is estimated at 27.1% [1]. Among the low and middle-income countries, the prevalence of LTBI in Kenya [63], Zimbabwe [64], Russia [65], Brazil [66], Vietnam [67], Rwanda [68], China [69] and South Africa [70] has been higher than in Iran [1]. The prevalence of LTBI in HCWs of Italy [71], Norway [72] and India [73] is reported to be equal to or less than Iran. Iran is a TB endemic country [25] and the treatment of LTBI is usually done by using a single medicine and only in high risk groups [74]. While in high-income countries, screening of pulmonary and lab staffs is recommended annually [75]. Also, it could be seen that training of Iranian's HCWs is not sufficient to prevent tuberculosis [25].

According to the results of meta-analysis, the lowest prevalence of LTBI among Iranian's HCWs was in southern Iran (18.2%). The highest prevalence of LTBI among Iranian's HCWs was reported in northern and western Iran. The high prevalence of LTBI among Iranian's HCWs may be due to neighboring Azerbaijan and Iraq [76]. Azerbaijan which is listed on the high burden countries has high prevalence of multidrug resistance MTB [76–78]. In fact, the northern neighbors of Iran, such as Kazakhstan, Azerbaijan, are among the high burden countries with a high prevalence of multi-drug resistant tuberculosis [80]. On the other hand, the name of the country's western neighbor of Iran–Iraq is not listed on the high burden countries [76–78] but according to reports from Ministry of Health—Iran Center for Medical Education and Treatment, Infectious Disease Control Center- the Iraqi state may have become a high-risk source for tuberculosis after undergoing its recent crisis [79, 80].

The current study showed that 15.5% of the Iranian's HCWs had used before the BCG with at least a positive PPD test. According to studies, BCG does not protect adults from getting infected with tuberculosis, so the positive results of tuberculin testing in people vaccinated with BCG will be considered as a latent infection [81]. In other words, previous vaccination with BCG prevents tuberculin testing [82]. This may be due to a false positive reaction in PPD [25]. Iranian's HCWs may respond to skin tests without being infected with mycobacterium [83]. The reason for these false-positive reactions may be due to contamination with non-*tuberculosis mycobacterium*, previous BCG, poor test performance or inappropriate interpretation of the test [83].

## 5. Limitations

Information about this meta-analysis was extracted from data published in Iranian databases as there was no access to the actual information of the control center of the Ministry of Health and Medical Education, so the exact prevalence of LTBI among Iranian's HCWs could not be calculated. Selection bias is able to limit the generalization of these findings because the type of bacteria strains in a country could be different with the other countries and could be related to descent diversities.

On the other hands, patients may not respond to skin test tuberculosis, even if they are infected with *Mycobacterium*. It may be due to skin allergies, recent infections (recent contact for 8 to 10 weeks), chronic infection, recent vaccinations with live viruses, advanced tuberculosis, some viral diseases (measles and bile), misdiagnosis skin or incorrect interpretation of the

reaction. Patients may also respond to skin tests, even without being infected with *Mycobacterium*. The reason for these reactions may be due to contamination with non-tuberculosis *Mycobacterium*, previous BCG, inappropriate test run or inappropriate interpretation of the test.

Despite the fact that the CDC updates the guidelines for the prevention and transmission of *M. tuberculosis* in health-care settings annually, the protocol for among Iranian's HCWs has not yet been prepared. Also, workshops could be developed to train tuberculosis prevention and self-care among Iranian's HCWs in the western regions.

National databases are not sensitive to operators "AND" and "OR" to search for the combinations. Also, some databases were not fully accessible because of using Guilan University of Medical Sciences'—Iran Ministry of Health & Medical Education- VPN.

## 6. Conclusion

This meta-analysis showed the prevalence of LTBI among Iranian's HCWs and estimated at 27.1%. The prevalence of LTBI between HCWs of Italy, Norway and India is reported to be equal to or less than Iranian's HCWs. On the other hand, the highest prevalence of LTBI among Iranian's HCWs in the north and the west of Iran may due to neighboring with Azerbaijan and Iraq which has become a high-risk source for tuberculosis by overcoming its recent years of crisis. Meanwhile, it could be seen that training of Iranian's HCWs is not sufficient to prevent tuberculosis. We also found that BCG was not able to protect Iranian's HCWs from TB infectious, completely.

## Supporting information

**S1 File. PRISMA checklist.**
(DOC)

**S2 File. The review protocol which has been registered in PROSPERO international prospective register of systematic reviews.**
(PDF)

**S3 File. Newcastle-Ottawa scale checklist.**
(PDF)

**S1 Table. Data characteristics (full details) (Microsoft excel).**
(XLSX)

**S1 Fig. The prevalence subgroup analysis of TST/PPD induration diameter (48 hrs.) in HCWs with LTBI (forest plot—Random effect model).**
(TIF)

**S2 Fig. The prevalence subgroup analysis of TST/PPD induration diameter (48 hrs.) in HCWs with LTBI base on geographical region (forest plot—Random effect model).**
(TIF)

**S3 Fig. Cumulative meta-analysis for overall prevalence of LTBI in HCWs.**
(TIF)

**S4 Fig. The sub group analysis of the quality of studies.**
(TIF)

**S5 Fig. The prevalence subgroup analysis to employment duration base on PPD (A), and QFT (B) in HCWs with LTBI (forest plot—Random effect model).**
(TIF)

**S6 Fig. The prevalence subgroup analysis of gender base on PPD (A), and QFT (B) in HCWs with LTBI (forest plot—Random effect model).**
(TIF)

**S7 Fig. The prevalence subgroup analysis of age base on PPD (A), and QFT (B) in HCWs with LTBI (forest plot—Random effect model).**
(TIF)

**S8 Fig. The prevalence subgroup analysis of history TB contact base on PPD (A), and QFT (B) in HCWs with LTBI (forest plot—Random effect model).**
(TIF)

**S9 Fig. The prevalence subgroup analysis of TB clinical symptoms base on PPD (A), and QFT (B) in HCWs with LTBI (forest plot—Random effect model).**
(TIF)

**S10 Fig. The prevalence subgroup analysis of BCG base on PPD (A), and QFT (B) in HCWs with LTBI (forest plot—Random effect model).**
(TIF)

**S11 Fig. Publication bias of studies included due to the aim of prevalence of HCWs with LTBI.**
(TIF)

## Acknowledgments

We would like to thank the vice chancellor of research and technology, Guilan University of Medical Sciences (ID: IR.GUMS.REC.1397.513).

## Appendix: PubMed search strategy

((Latent Tuberculosis) AND Iran AND Prevalence AND ((Health Personnel) OR (Healthcare Worker) OR (Health Care Provider)).

## Author Contributions

**Conceptualization:** Mohammad Hossein YektaKooshali, Alireza Jafari.

**Data curation:** Mohammad Hossein YektaKooshali.

**Formal analysis:** Mohammad Hossein YektaKooshali.

**Funding acquisition:** Mohammad Hossein YektaKooshali.

**Investigation:** Mohammad Hossein YektaKooshali, Farahnaz Movahedzadeh, Ali Alavi Foumani, Alireza Jafari.

**Methodology:** Mohammad Hossein YektaKooshali.

**Project administration:** Mohammad Hossein YektaKooshali, Alireza Jafari.

**Resources:** Alireza Jafari.

**Software:** Mohammad Hossein YektaKooshali.

**Supervision:** Farahnaz Movahedzadeh, Ali Alavi Foumani.

**Validation:** Mohammad Hossein YektaKooshali.

**Visualization:** Mohammad Hossein YektaKooshali.

**Writing – original draft:** Mohammad Hossein YektaKooshali, Hoda Sabati, Alireza Jafari.

**Writing – review & editing:** Mohammad Hossein YektaKooshali, Hoda Sabati, Alireza Jafari.

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
