## [Decision Letter · Decision Letter 0]

30 Aug 2019

[EXSCINDED]

PONE-D-19-18112

Is latent tuberculosis infection challenging in Iranian health care workers? A systematic review and meta-analysis

PLOS ONE

Dear Dr. Jafari,

Thank you for submitting your manuscript to PLOS ONE. After careful consideration, we feel that it has merit but does not fully meet PLOS ONE’s publication criteria as it currently stands. Therefore, we invite you to submit a revised version of the manuscript that addresses the points raised during the review process.

We would appreciate receiving your revised manuscript by Oct 14 2019 11:59PM. To enhance the reproducibility of your results, we recommend that if applicable you deposit your laboratory protocols in protocols.io, where a protocol can be assigned its own identifier (DOI) such that it can be cited independently in the future. For instructions see: http://journals.plos.org/plosone/s/submission-guidelines#loc-laboratory-protocols

We look forward to receiving your revised manuscript.

Kind regards,

HASNAIN SEYED EHTESHAM

Academic Editor

PLOS ONE

Additional Editor Comments:

Major Revision

3. We note that Figure 5 in your submission contain a map image which may be copyrighted.

a. You may seek permission from the original copyright holder of Figure  to publish the content specifically under the CC BY 4.0 license. 

Reviewers' comments:

Reviewer's Responses to Questions

**Comments to the Author**

1. Is the manuscript technically sound, and do the data support the conclusions?

Reviewer #1: Partly

Reviewer #2: Partly

2. Has the statistical analysis been performed appropriately and rigorously? 

Reviewer #1: Yes

Reviewer #2: No

3. Have the authors made all data underlying the findings in their manuscript fully available?

Reviewer #1: Yes

Reviewer #2: No

4. Is the manuscript presented in an intelligible fashion and written in standard English?

Reviewer #1: No

Reviewer #2: No

5. Review Comments to the Author

Reviewer #1: Health care workers (HCWs) are at increased risk of latent tuberculosis infection (LTBI). This study is aimed at conducting a systematic review and meta-analysis of the prevalence and incidence of LTBI in Iranian HCWs. English throughout the manuscript is flawed and makes it a difficult read. However authors have done meticulous meta-analysis. Portions which require correction have been highlighted in yellow in the attached pdf.

The whole manuscript needs to be revamped majorly and revised then based on scientific criteria.

Specific comments are given below

1. Abstract: Background section line 28 “changed” does not make any sense

2. Was there any period from when studies were collected?

3. Line 64, provide the range of prevalence in other reports

4. Line 69, change “transfer” to “spread”

5. Mention the number of low, medium and high quality studies

6. Line 190, “it was considered significant at p<0.05

7. Line 234-239 not clear

8. Line 276, add p value

9. Line 308, rephrase

10. Did the prevalence estimates correlate to WHO incidence rates?

11. Legends for all figures need to be improved with the aim of giving correct aim of the experiment and its analysis

12. Line 263 says “there was no significant relationship” and then gives p<0.0001 which is contradictory

13. What kind of studies have been included cohort/cross sectional/ case studies?

14. In methodology section, which software package was used for sensitivity test?

15. It is suggested that a more critical account be provided in the discussion section taking into account salient findings, problems and gaps

16. Check the entire MS for repetitions

Reviewer #2: The meta-analysis performed in the manuscript shows the highest prevalence of Latent TB infection in health care workers in the north and the west of Iran due to neighboring countries like Azerbaijan and Iraq, respectively. Hence, there is a need to create awareness in Iranian HCWs about isolation and personal protection.

however the study handled a very serious issue but following points need to be addressed

1. Please follow the proper PRISMA template and not a Venn Diagram

2. What is the basis of sensitivity analysis is not clear. What is the rationale for removing that one study?

3. The S2 figure is more important than the Figure 5 with map. Maybe the map can go in supplementary

4 As they say national databases are not sensitive to Boolean operators, do the authors think this affects the search? Does this mean it is not an exhaustive search on this topic?

5. Doing a sensitivity analysis based on the quality of study will also add value.

6. The search strategy provided by the authors is very very broad. They do not include the tests, they do not include different terms for HCWs, eg: T-SPOT, community workers, TST etc. I do not feel this is a complete and detailed search.

7. Was data extracted for both one-step and two-step TST? If so, doing that analysis is also important. The authors say they did it but is not available.

6. PLOS authors have the option to publish the peer review history of their article (what does this mean?). If published, this will include your full peer review and any attached files.

Reviewer #1: No

Reviewer #2: No

---

## [Author Response · Author response to Decision Letter 0]

15 Sep 2019

Is latent tuberculosis infection challenging in Iranian health care workers? A systematic review and meta-analysis

Manuscript ID: PONE-D-19-18112

Editor Recommendations:

Comment: It was checked.

Comment: The English language, spelling, and grammar of manuscript was rechecked by Dr. Farahnaz Movahedzadeh as a Research Assistance Professor of UIC. 

3. We note that Figure 5 in your submission contain a map image which may be copyrighted. 

Comment: The map image which we used in this manuscript depicted by authors and we didn’t copy from other sources. we are ready to change map if you want.

Reviewer #1 recommendations: 

Health care workers (HCWs) are at ‎increased risk of latent tuberculosis infection ‎‎(LTBI). This study is aimed at conducting a ‎systematic review and meta-analysis of the ‎prevalence and incidence of LTBI in Iranian ‎HCWs. English throughout the manuscript is ‎flawed and makes it a difficult read. However, ‎authors have done meticulous meta-analysis. ‎Portions which require correction have been ‎highlighted in yellow in the attached pdf. The whole manuscript needs to be revamped ‎majorly and revised then based on scientific ‎criteria. 

Comment: Background section line 28 ‎‎“changed” does not make any sense. 

Response: The background section line 28 was improved to “The high chances of getting latent tuberculosis infection (LTBI) among health care workers (HCWs) will an enormous problem in low and upper-middle-income countries.”

Comment: Was there any period from when studies ‎were collected? 

‎Response: Absolutely yes. The time interval of this study was without any time limitation till January 01, 2019.

Comment: Line 64, provide the range of prevalence in ‎other reports 

‎Response: It was improved to “There are several reports of TB outbreaks in Iran. According to the Iranian’s ministry of health, the incidence and the prevalence of TB are high in Sistan and Baluchestan, Khorasan, Mazandaran, Guilan, West and East Azerbaijan, Ardabil, Kurdistan, Khuzestan and southern coasts. Conversely, the incidence and the prevalence of TB are low in the central parts of Iran. The highest incidence and prevalence of TB belong to Golestan and Sistan-Baluchistan.” 

Comment: Line 69, change “transfer” to “spread” 

‎Response: It was done. 

Comment: Mention the number of low, medium and ‎high quality studies 

‎Response: The quality scores of studies were mentioned in Table S1. 

Comment: Line 190, “it was considered significant at ‎p<0.05 

‎Response: It was done. 

Comment: Line 234-239 not clear 

‎Response: It was improved to ” In the PPD test, the prevalence of LTBI in Iranian’s HCWs with more than 10 years old work-experience was evaluated 51%. The prevalence of LTBI in Iranian’s HCWs with less than 10 years old work-experience was estimated at 29.30%. In the QFT test, the prevalence of LTBI in Iranian’s HCWs with more than 20 years old work-experience was calculated 20.49% [CI95%: 11-34.97], which showed a significant relationship between the duration of employment (P <0.0001) (S4 Fig)”. 

Comment: Line 276, add p value 

‎Response: It was done.

Comment: Line 308, rephrase 

‎Response: It was done.

Comment: Did the prevalence estimates correlate to ‎WHO incidence rates? 

‎Response: In this study, we just estimated the prevalence of LTBI among Iranian HCWs, not the incidence. 

Comment: Legends for all figures need to be ‎improved with the aim of giving correct aim of ‎the experiment and its analysis 

‎Response: It was done.

Comment: Line 263 says “there was no significant ‎relationship” and then gives p<0.0001 which is ‎contradictory 

‎Response: It was improved to “The highest prevalence of LTBI in Iranian HCWs aged 30 years old was estimated 22.52% [CI95 %: 3.7-68.34] in the QFT. In both PPD test and QFT, it was evaluated that there was significant relationship between the prevalence of LTBI in HCWs, and age of HCWs (P<0.0001) (S6 Fig).”

Comment: What kind of studies have been included ‎cohort/cross sectional/ case studies? 

‎Response: The cross sectional and cohort studies have been included. 

Comment: In methodology section, which software ‎package was used for sensitivity test? 

‎Response: Comprehensive Meta-Analysis (Ver 2 , Englewood, NJ 07631, USA)‏.

Comment: It is suggested that a more critical account ‎be provided in the discussion section taking ‎into account salient findings, problems and ‎gaps 

‎Response: In light of the evidence, we preferred to focus more on the prevalence of LTBI among HCWs in the eastern provinces of Iran, specially provinces which neighbor of Iraq. 

Comment: Check the entire MS for repetitions 

‎Response: It was done.

 

Reviewer #2 recommendations:

The meta-analysis performed in ‎the manuscript shows the highest prevalence ‎of Latent TB infection in health care workers in ‎the north and the west of Iran due to ‎neighboring countries like Azerbaijan and Iraq, ‎respectively. Hence, there is a need to create ‎awareness in Iranian HCWs about isolation and ‎personal protection. However, the study handled a very serious issue ‎but following points need to be addressed 

‎ Comment: Please follow the proper PRISMA template ‎and not a Venn Diagram 

‎Response: It was done.

‎ Comment: What is the basis of sensitivity analysis is ‎not clear. What is the rationale for removing ‎that one study? 

‎‎Response: According to the Cochrane hand book, by creating a given set of variables, an analyst can determine how changes in one variable affect the outcome. In other words, Sensitivity analysis is a method for predicting the outcome of a decision if a situation turns out to be different compared to the key predictions. It helps in assessing the riskiness of a strategy. Helps in identifying how dependent the output is on a particular input value.

https://handbook-5-1.cochrane.org/chapter_9/9_7_sensitivity_analyses.htm

It used a lot in other same meta-analysis which is published.

https://www.sciencedirect.com/science/article/abs/pii/S1871402119303960

https://journals.plos.org/plosone/article?id=10.1371/journal.pone.0214738

https://journals.plos.org/plosone/article?id=10.1371/journal.pone.0164769

http://ijn.mums.ac.ir/article_13419.html

https://www.ncbi.nlm.nih.gov/pubmed/30573555

Comment: The S2 figure is more important than the ‎Figure 5 with map. Maybe the map can go in ‎supplementary 

‎‎Response: We appreciate your valuable comment, but we believe that the map could be more clearer and it could convey the author's intent to the readers.

Comment: As they say national databases are not ‎sensitive to Boolean operators, do the authors ‎think this affects the search? Does this mean it ‎is not an exhaustive search on this topic? 

‎Response: Thank you for your attention. In national databases, the SID is ‎ just sensitive to Boolean operators. So, we searched other Persian national databases (Barakat knowledge network system; Irandoc, Magiran, and Iranian national library) through manual ways.

Comment: Doing a sensitivity analysis based on the ‎quality of study will also add value. 

‎‎Response: The sub group analysis of the ‎quality of studies was showed in (S4 Fig).

Comment: The search strategy provided by the authors ‎is very broad. They do not include the ‎tests, they do not include different terms for ‎HCWs, eg: T-SPOT, community workers, TST ‎etc. I do not feel this is a complete and ‎detailed search. 

Response: Thanks to your valuable comment. To including of different terms of LTBI in Iranian ‘s HCWs. we searched more synonymous of keywords in the national databases. 

Comment: Was data extracted for both one-step and ‎two-step TST? If so, doing that analysis is also ‎important. The authors say they did it but is not ‎available. ‎

‎ Response: As shown in Table S1, only two studies extracted one-step and two-step TST data that could not be analyzed.

---

## [Editor Report · Decision Letter 1]

19 Sep 2019

Is latent tuberculosis infection challenging in Iranian health care workers? A systematic review and meta-analysis

PONE-D-19-18112R1

Dear Dr. Jafari,

We are pleased to inform you that your manuscript has been judged scientifically suitable for publication and will be formally accepted for publication once it complies with all outstanding technical requirements.

With kind regards,

HASNAIN SEYED EHTESHAM

Academic Editor

PLOS ONE

Additional Editor Comments (optional):

The Authors have very comprehensively revised the manuscript addressing every single issue raised by the 2 Reviewers. The important issues such as performing a sensitivity analysis and other issues have now been take care.

I recommend publication of this manuscript.
---

## [Editor Report · Acceptance letter]

24 Sep 2019

PONE-D-19-18112R1 

Is latent tuberculosis infection challenging in Iranian health care workers? A systematic review and meta-analysis 

Dear Dr. Jafari:

I am pleased to inform you that your manuscript has been deemed suitable for publication in PLOS ONE. Congratulations! Your manuscript is now with our production department. 

With kind regards,

on behalf of

Prof HASNAIN SEYED EHTESHAM 

Academic Editor

PLOS ONE